# Experimental Field Tests of the Suitability of a New Seeder for the Soils of Northern Kazakhstan

**Mubarak Aduov [1], Saule Nukusheva [1], Talgat Tulegenov [1], Kadirbek Volodya [1], Kanat Uteulov [1], Bolesław Karwat [2] and Michał Bembenek [2,*]**

1   Department of "Agricultural Engineering and Technology", Technical Faculty, NJSC "Saken Seifullin Kazakh Agrotechnical Research University", Zhenis Ave. 62, Astana 010011, Kazakhstan
2   Faculty of Mechanical Engineering and Robotics, AGH University of Science and Technology, A. Mickiewicza 30, 30-059 Krakow, Poland
*   Correspondence: bembenek@agh.edu.pl

**Abstract:** Kazakhstan is historically a livestock country, and the production of feed requires no less attention than the production of grain. To improve the forage base, one solution is the sowing of high-yielding fodder seeds. An experimental seeder was developed with new design solutions for the sowing machine, with three blades installed at an angle of 120° relative to the lower part of the blower shaft, deviated vertically by 8–10°, along with components with a radius vector of 10–15° and the blower shaft attached to the top of the sowing cylinder. The closing part of the disc coulter contained the press rollers with a disc diameter measuring 350 mm. The field tests were conducted with the parameters between the discs set to $\alpha = 10°$, a disc vanishing point of $\beta = 40°$, a coulter angle of 32°, and an individual 320-mm press roller with a cylindrical 60-mm rim, a leash, and a section for setting the seed placement depth. The wheatgrass varieties "Burabay" and awnless brome "Akmola emerald" were sown. The research showed the higher efficiency of the experimental seeder with seeding units and sowing parts compared to a serial seeder in terms of agricultural performance. The increase in seed germination was 3.56%. The experimental seeder surpassed the regular seeder by 4.95% in terms of the depth uniformity of the seed placement, in terms of yield increase by 5.361 cwt/ha, with reductions in traction resistance of 12.3%, and in fuel consumption by 10%. The economic efficiency from the fuel reduction and yield increase was estimated at around 7700 USD/ha per year.

**Keywords:** mounted seeder; seeding unit; ridge breaker; non-flowing seeds; traction resistance; sowing depth

## 1. Introduction

Kazakhstan is a country with an area of 190 million ha. According to the Kazakhstan Land Resources Agency, it contains 188 million ha of natural forage land [1]. Problems in rangeland farming have been studied by scientists from various countries in different ways [2–4]. To improve the forage base, one solution is to sow high-yielding seeds of fodder crops [5–7]. This problem is so urgent that scientists, breeders [6–8], agronomists [9–11], and agricultural equipment engineers [12–14] are working to solve it [9,15,16]. The development of seeders with new working tools, e.g., coulters, was conducted by scientists from different countries [17–19]. In addition, the application of new technologies requires the development of new technical solutions capable of high-quality seeding [15,20,21] and increasing yields [22]. Studies on the development and justification of technological parameters and the testing of seeders for the cultivation of grain crops have also been conducted by scientists from Kazakhstan [23].

A research and production experience analysis shows that the actual state of the fodder base can be found on the pages of "KazakhZerno.kz" [24] and in recommendations by employees of LLP, "NPZH, named after A.I. Barayev." According to A.I. Barayev, the fodder base does not meet the needs of animals. Therefore, fodder production requires the same

attention as grain production. Although significant research has been conducted on the development of cultivation machines [25–27], unfortunately, there is a large gap in terms of the research into the development of versatile seed drills with automatically variable designs and the adjustment of their technological parameters. These can be used in different climatic zones with different soil characteristics [28–30]. An analysis of the scientific literature shows an urgent need for the development of new devices designed for seed sowing [31–33]. One challenge in the development of new seed drills is the investigation of the influence of sowing principles on the germination of seeds in mixed grassland [34–36]. The effects of improved seed drills, which implement different seeding principles, on seeding quality, seed germination, and yields are also under study [37–39]. Another task in the development of new seed drills is the improvement of seed germination by applying the best possible seed-drop height and tractor speed [40,41]. In addition, attention is focused on increasing the seeding rate in fields through improved seed drills [42,43].

Only cooperation between science, production, the development of new technologies, and recommendations can solve the above-mentioned problems. However, obtaining high yields of fodder crops directly depends on both sowing methods and the implementation and adoption of these technologies.

According to Zhulmukhametova's speech at the Kazakhstan Ministry of Agriculture in 2017, the decisive factor in the sustainable development of livestock is the provision of fodder. However, the country is facing of a field-unit deficit of 8.7 million tons, which is the reason for the 13% shortfall in gross livestock production. To reduce this deficit by 2025, specific targets have been set using global experience and rapidly implementing it in agriculture [44]. Kazakh scientists drafted a program for developing fodder production, which considers the peculiarities of regions and makes calculations on the content of fodder units, depending on the nutritional value of the fodder crops produced in the area [45]. Furthermore, recommendations are prepared based on the structures of sown areas and by considering the natural and climatic characteristics of regions. In the northern part of Kazakhstan, between 60% and 70% of grain sowing and forage crops are obtained through SZS-2.0, SZTS-6, and SZTS-12 seeders–cultivators [46]. The rest of the acreage is sown by seeders like SKP-2,1 as well as foreign seeders, like those manufactured by John-Deere, Horsch, Argentinian Crucianelli, Pionera Flexi-Coil, Concord, and Amazonia, etc., or readjusted seeders. Furthermore, the domestic seed-drill park is not being renewed, and Kazakhstan has become completely dependent on foreign agricultural equipment suppliers. It is not possible to purchase expensive equipment for small companies, at least for medium-sized farmers. Moreover, the application of new technologies is not possible without the appropriate technical support. Currently, in Kazakhstan, it is necessary to modernize the existing and create new equipment, in light of domestic and foreign experience. Considering the zonal conditions of northern Kazakhstan, imported seed drills do not always correspond to the catalogue indicators [47].

The problems with forage shortages discussed in this article can be solved by creating new high-tech seeding machines. The advantage of the newly developed seeder is the completely new design of the sowing unit, which is able to evenly distribute seeds over areas with small depths of placement of non-flowing grass seeds without injuring them. The literature review established that only disc coulters provide the necessary uniformity of non-flowing grass seeds at a shallow placement depth. Additionally, they provide low traction resistance.

Drawing upon an analysis of the coulters found in current grass seeders and recognizing the limitations within the coulter's technological process, along with considerations for the distinctive physical and mechanical properties of the soil in Northern Kazakhstan, the authors formulated and put forth the design and technological parameters for enhancing the closing component of the disc coulter seeder. However, the analysis indicates that the primary drawback observed after seed sowing is the inadequate performance in rolling the seeds. This results in uneven and insufficient soil compaction both on the sides and above the seeds. An investigation into the operation of the rollers within the sowing sections

of seeders with various designs has led to the conclusion that soil compaction quality is influenced by its physical and mechanical properties. This dependence extends to the specific pressure, speed, and design parameters of the rollers. To address this issue, an individual press roller was developed. This roller features a cylindrical rim, a leash, and a sector for adjusting the depth of seed placement. The seeder has a grip width of 3.60 m, row spacing of 0.30 m, and a seeding rate ranging from 10.0 to 30.0 kg/ha, with a seed placement depth of 2.0–8.0 cm. Adhering to international standards, a 350 mm diameter disk was proposed with recommended parameters: $\alpha = 10°$, $\beta = 40°$ for the vanishing point of the disk, and $\alpha = 32°$ for the coulter angle. Additionally, a press wheel with a diameter of 320.0 mm and a rim width of 60.0 mm was introduced. The proposed configuration, with specified parameters between the discs and the coulter angle, enhances the uniform distribution of seeds across the row's width. It also optimally positions the seeds at the bottom of the furrow with a consistent seeding depth, facilitated by the corresponding wheel diameter for effective seed rolling. The disc travel distance is $\pm 10$ cm. Employing the suggested equipment enables more efficient seed rolling, establishing firm contact between seeds and soil. This practice, in turn, enhances plant development, growth, and ultimately, increases the yield of grain and forage crops.

The newly developed seeder underwent testing using the wheatgrass varieties "Burabay" and the awnless brome "Akmola emerald." These outcomes were subsequently juxtaposed with tests conducted using a series seed drill. Finally, an economic evaluation was performed, considering the results obtained during the sowing process in relation to both yield augmentation and reductions in fuel and lubricant costs.

## 2. Materials and Methods

To increase the indicators of quality for the seeding of non-flowing seeds, the authors substantiated and developed a grass seeder equipped in a technological process controller. The novelty of the developed seeder is confirmed by patents and applications for intellectual property (Aduov et al., 2020, 2015, 2016 (patents)) [22,43]. The detailed description of the seeder, the improved seeding unit, the seeding unit with an intelligent control unit, and the entire technological process of seeding are presented in articles [21,23].

The improved seeder for non-flowing grass seeds (Figure 1) consists of the main frame (1) and auxiliary bars (2). Two pneumatic wheels (7) are installed on the main bar of the frame (1) with an automatic hitch (3). Twelve sowing sections are pivotally attached to the rear part of the frame (4). The sowing section is folded from a double-disk coulter (5) and a roller (6). Six seed hoppers (8) are rigidly connected to the sowing machines (9), and each one has two distribution sleeves (10), connected to the sowing section via a seed tube (11).

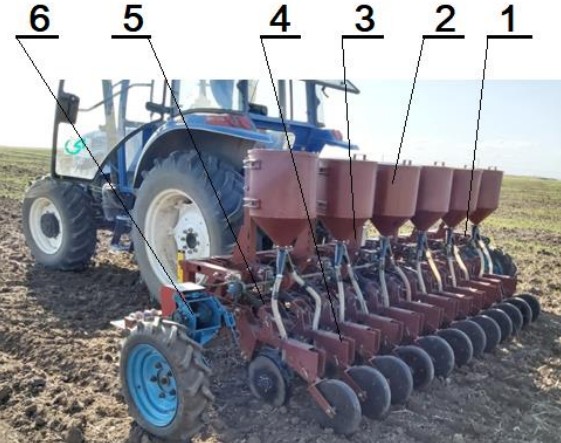

**Figure 1.** The view of the experimental seeder for sowing non-flowing grass seeds: 1—frame; 2—seed box with sowing machines; 3—hinged device; 4—sowing section; 5—seeding machine drive; 6—running gear.

The prototype seeder with an intelligent control unit of the technological process for sowing non-flowing grass seeds is characterized by: a working width of 3.6 m, a width between the rows of 0.30 m, a seeding rate range from 10.0 to 30.0 kg/ha, and a seeding depth of 2.0 to 8.0 cm.

- Based upon theoretical studies of the technological process of the proposed seeding unit, the analytical relationships of seed movement at various stages in the seeding unit were obtained. This analysis facilitated the identification of optimal values of the technological and design parameters of the device. Those values ensured high-quality non-flowing grass seed sowing based on the following: the height of the tedder should be from 6 to 8 mm;
- the angle of the vertical blades from 8 to 10°;
- the angle of the vector radius from 10 to 15°; and
- the helix angle $\alpha$ must not exceed 17° and the radius must be within 0.02 m.

The structural dimensions of the distributor's head were considered based on the screw dimensions and the coefficient of the friction of non-flowing seed on steel.

When sowing, the main task is to distribute the seed evenly over the area.

The international standard discs of diameter at 350 +/− 5 mm were used. Recommended parameters are as follows: disc angle $\alpha = 10°$, disc vanishing point position $\beta = 40°$, and coulter arm angle $\gamma = 32°$.

The diameter of the press wheel is determined with Equation (1) as follows:

$$D \le \frac{2 \cdot h}{1 - cos\alpha} \tag{1}$$

where $h$ is the depth of the track, mm, and $\alpha$ is the angle of the roller rim around the soil ($\alpha = 15$ to 20°).

The following recommendations were used when selecting and dimensioning the coulter bar and the coulter bar with packer roller:

- the lifting height of the coulters was assumed to be equal to the maximum seeding depth plus 6 to 7 cm for adaptation to the field topography; and
- the force on the lever for lifting and moving coulters to the working position must not exceed 196.2 N.

Considering the technological and physical-mechanical properties of soil and seeds of cultivated crops and the analysis of kinematics and forces of the sowing mechanism, the following design dimensions of the coulter and packer roller are determined [33]:

- the disc diameter at 350 mm;
- the angle between the discs at $\alpha = 10°$;
- the disc vanishing point position at $\beta = 40°$;
- the coulter angle at $\gamma = 32°$; and
- the packer roller at a diameter of 320 mm and a rim width of 60 mm.

The intrinsic flow rate of hard-to-loose seeds of forage grasses without vibration ranges from 0.116 to 0.176 m/s, which is not enough for them to flow out of the hopper with no external force. In this connection, a new sowing apparatus with three blades was created to diminish the resulting arches of the sown material. The blower shaft (Figures 2 and 3) is attached to the top of the sowing cylinder via a threaded connection. See also Figure 4.

The control unit includes an electronic control unit, a path-detecting sensor, a seeding control sensor, and the hopper level sensor. The data from all the sensors can be presented on the tractor display (Figure 5).

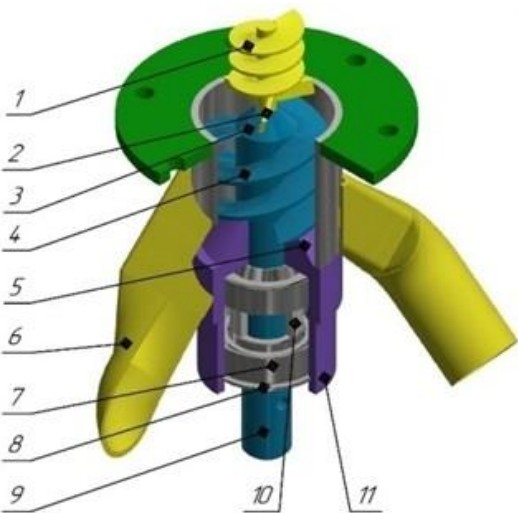

**Figure 2.** The scheme of a sowing unit for non-flowing grass seeds: 1—rotator; 2—blade supercharger; 3—cone; 4—helical spiral; 5—lower cone; 6—sleeve; 7—bearings; 8—retaining rings; 9—shaft; 10—intermediate ring; 11—bunker.

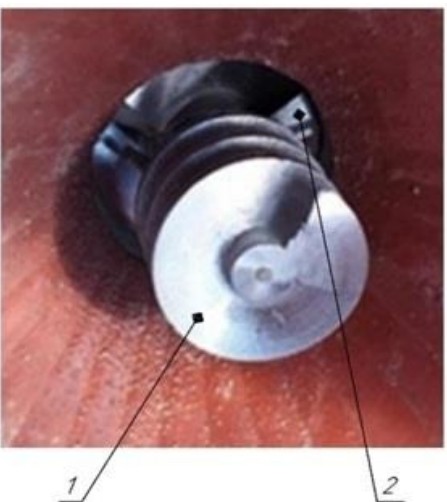

**Figure 3.** The image of a sowing unit for non-flowing grass seeds: 1—rotator; 2—blade supercharger.

The bench tests were conducted to objectively assess the technical level and quality of manufacturing of the experimental seeder. They showed the following:

- the experimental seeder sowing capacity was 8 to 30 kg/ha;
- the uneven seeding rate was 4.6% for wheatgrass and 4.8% for awnless bromegrass;
- the total seeding instability was 2.9% for wheatgrass and 2.7% for awnless bromegrass; and
- seed crushing was 0.1%.

Thus, the bench test results of the seeder experimental prototype met the technical specification requirements and Kazakhstan state standards for seeding machines.

For laboratory and field seeder tests, the following was determined: a driving speed of 5, 7, and 9 km/h and a seed sowing depth of 2 to 5 cm. The spacing between rows was 30 cm, and the rate of seeding was within the minimal and maximal values for the chosen crop's range.

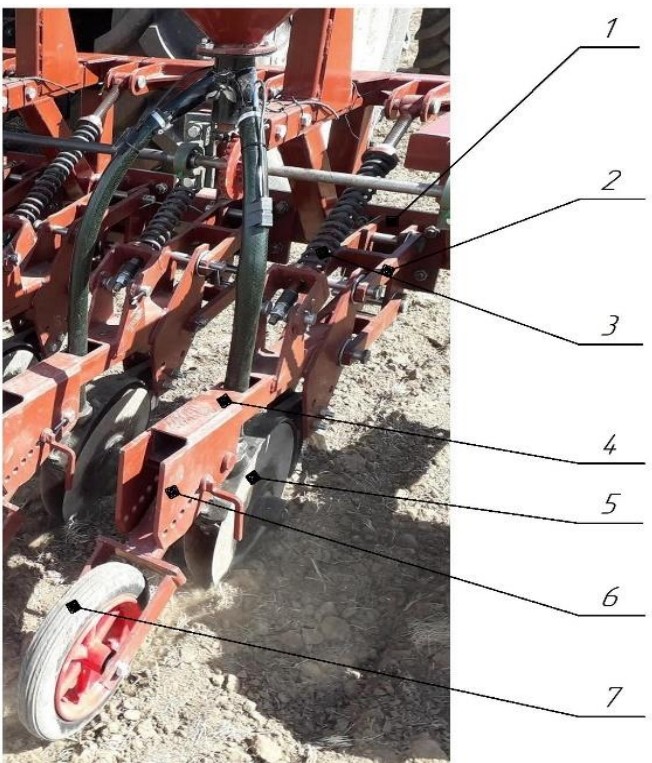

**Figure 4.** Closing part of the seeder: 1—bracket; 2—parallelogram mechanism; 3—rod with a spring; 4—longitudinal beam; 5—closing disc; 6—sector; 7—wheel.

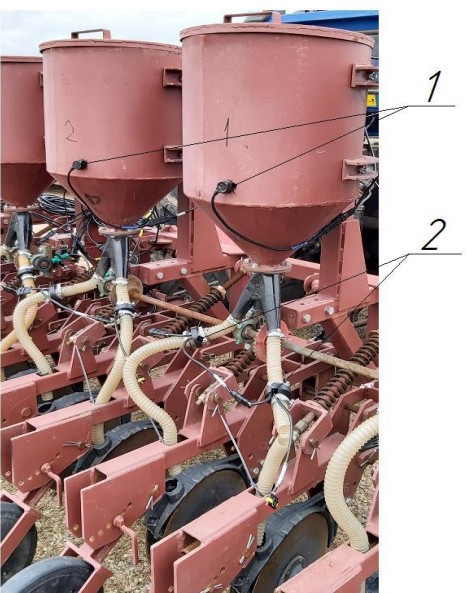

**Figure 5.** The placing of the sensors: 1—sensors for monitoring the level of the bunkers; 2—sensors for monitoring the sowing of the seed material.

The traction resistance determination of the experimental seeder was performed under the GOST20915-2011 and GOST31345-2007requirements.

The data collection and processing traction resistance and fuel consumption was performed with a IP 264 device manufactured by KubNIITiM (Passport UV 404176.029 PSOOO "Vector-PM, Russia Uralves K-R-20G-10t) manufactured on the individual order of the scientists from S. Seifullin Kazakh Agrotechnical University.

For data processing, Statistica (StatSoft) was used. For modelling and design, the Inventor (Autodesk) was performed.

Economic tests of the seed drill were carried out on a field with an area of 20 hectares in the farm "Guldana" Settlement: Yalta. District: Gabita Musrepova. Region: North Kazakhstan (52°27′21.6″ N 67°11′05.4″ E).

The experimental plot, measuring 150.0 m in length and 14.4 m in width, was situated within the premises of S. Seifullin Kazakh Agrotechnical University's research and production campus. Based on the results of the selection of perennial cereal grasses recommended by the Research and Production Center for Grain Farming A.I. Baraev, for cultivation in the conditions of Northern Kazakhstan, according to the growing season length, the yield of fodder mass and seeds, the quality of the fodder, and the resistance to adverse environmental factors, wheatgrass varieties "Burabay" (Figure 6) and awnless brome "Akmola emerald" (Figure 7) were sown. They are characterized by poor flowability, preventing high-quality sowing. In the pictures, it can be seen that the seeds of these crops are loaded into a pile or lump; since the seeds are not smooth, they have poor flowability.

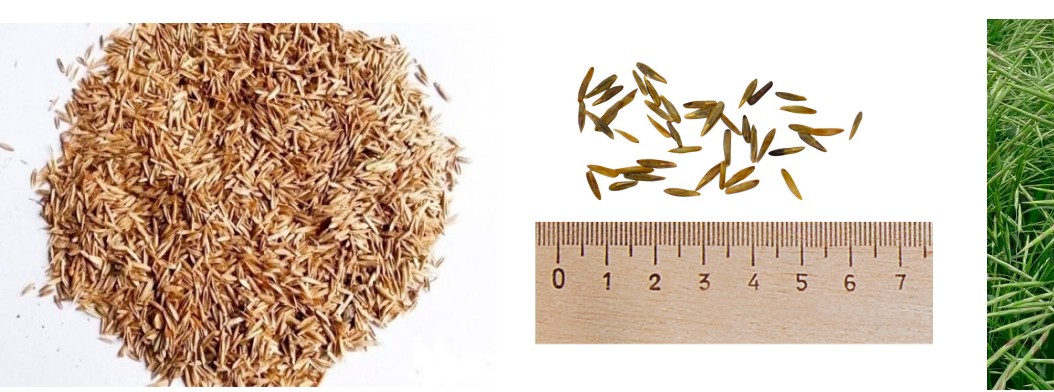
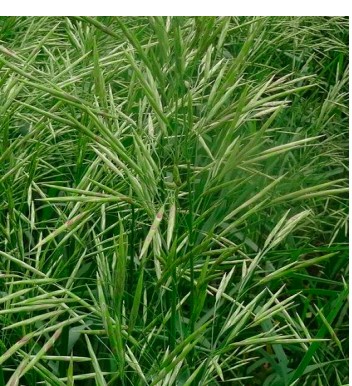

**Figure 6.** Images of the wheatgrass variety "Burabay".

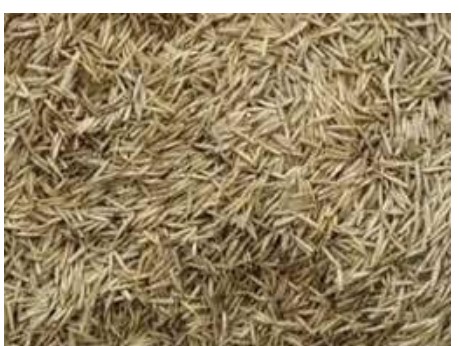
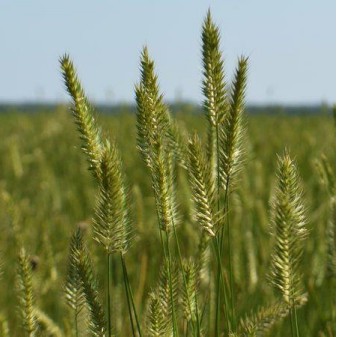

**Figure 7.** Images of the awnless brome "Akmola emerald".

The expected friction coefficient for loose seeds should be around 0.57, while for the wheatgrass variety "Burabay" it is 0.38 (which is 0.19 lower) and awnless brome "Akmola emerald" is 0.46 (which is 0.11 lower). The proper coefficient of internal friction for loose seeds is 0.97, while for wheatgrass variety "Burabay" is 2.18, which is 1.21 higher; for awnless brome "Akmola emerald" is 1.35, which is 0.38 higher. The value of the seed compaction coefficient ranges from 1.06 to 1.17, while for the wheatgrass variety "Burabay" it is 1.12 and for awnless brome "Akmola emerald" it is 1.17.

In addition, the zoned varieties of perennial grasses are highly productive, resistant to abiotic (frost, low temperatures in winter, and high temperatures in summer, etc.) and biotic (diseases and pests) environmental factors. Wheatgrass varieties "Burabay" and awnless brome "Akmola emerald" are able to provide hay yields of up to 30–45 cwt/ha, and with regular irrigation more than 60 cwt/ha. The seeds of these crops practically do

not crumble, and therefore the second advantage is longevity. The term of economic use of the steppe regions is from 4 to 6 years. However, these two crops exhibit poor flowability. This characteristic prompted the authors to select them for research purposes, conducting both laboratory and field tests. The seeding rate and the depth of seed placement were established based on the recommendations of the agronomist of the farm. In this case, the size of the experimental plot was 3.6 × 37 m. The collected phenological observations of the experimental plots were entered in the observation log.

For the agrotechnical and energy evaluation of the seeder under development, its traction resistance was previously determined in the analytical form presented by Aduov et al. [21,23]. The research program for the experimental seeder includes both the bench and laboratory field tests. The sown material prepared for testing was determined and evaluated based on its purity, damage, germination, moisture, and weight of 1000 seeds according to Gosudarstvennyy tandard (GOST). 20290-74 "Seeds of agricultural crops. Determination of sowing qualities of seeds. Terms and Definitions."

The performance quality of the seeder's technological process was evaluated at a speed of 7 km/h, and subsequently, it was enhanced by increasing the speed of the sowing machine by 25 to 30%. The tests were conducted at various speeds while maintaining the same settings of the sowing units.

The parameters of work quality during the sowing process of grass seeds with the new seeder for sowing non-loose grass seeds were compared with the standard SZT-3.6 (Astra, Ukraine) seeder, which until now sowed more than 50% of forage crops with loose and non-loose grass seeds from the main bunkers through disc coulters.

The unevenness and instability of seed sowing were determined with the economic sowing rate. The number of seeds that passed through sowing units (seed tubes) was determined by comparing the source material to the fragmented seeds. Tests were conducted using the Labor Code and GOST 26711-89 at the same working speed. Samples were collected at least three times.

## 3. Results and Discussion

An experimental model of a mounted seeder was developed and its main indicators were substantiated. Alongside the seeding depth, the design, and technological parameters of the seeding unit within the seeder, the authors also established the height of the ridge breaker for seeds within the hopper, the inclination angle of the blade concerning the vertical, and the angle of rise and the radius of the screw spiral. The test proved the corrected design and technological parameters of the seeding part of the seeder including the disc diameter, disc angle, position of a disc's convergence point, coulter leash angle, and diameter of the packer roller and its rim width. The experimental model of economic tests qualitatively seeded hardly friable and non-flowing seeds of fodder crops and provides the constant and unobstructed movement of seeds without piling up (and consequently clogging) the sowing part, and uniformly and favorably distributed seeds on the area in the soil for germination (Figure 8).

The results of the agronomic performance of the plots are shown in Tables 1–5.

**Table 1.** The results containing the density of planting.

| Type of Seeder | The Type of the Crop | Number of Plants, pcs/m$^2$ | Field Germination, % |
| --- | --- | --- | --- |
| Tractor HS1204 + seeder for sowing non-flowing grass seeds | wheatgrass variety "Burabay" | 295 | 90 |
| | awnless brome "Akmola emerald" | 331 | 89 |
| Tractor HS1204 + SZ-3.6 (Astra) | wheatgrass variety "Burabay" | 283.3 | 86.44 |
| | awnless brome "Akmola emerald" | 320.4 | 86.15 |

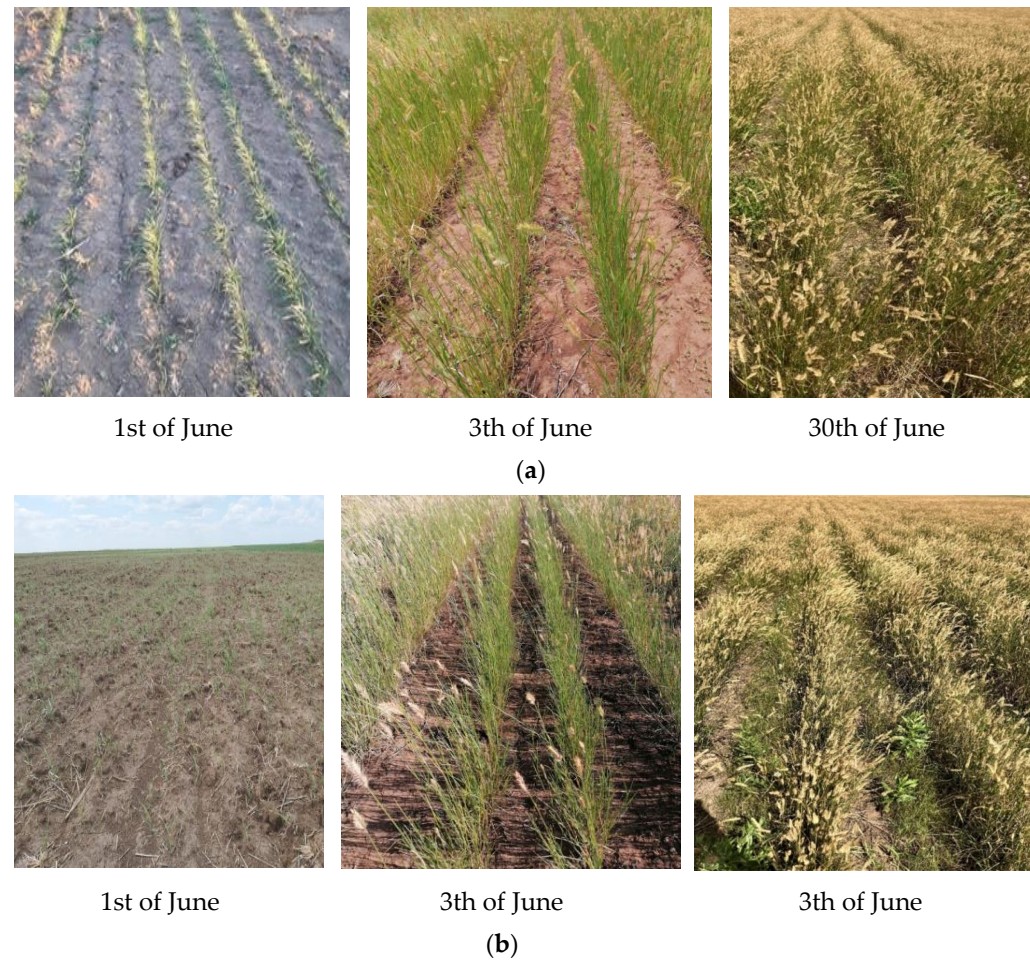

<div align="center">

1st of June      3th of June      30th of June

(**a**)

1st of June      3th of June      3th of June

(**b**)

</div>

**Figure 8.** Planting of Burabay wheat grass: (**a**) Crops of wheatgrass "Burabay" on the experimental plot. (**b**) Crops of wheatgrass "Burabay" in the control plot.

**Table 2.** The comparative performance indicators of the experimental and serial SZT-3.6 (Astra) seeders when sowing grass seed.

| Name of Indicators | Experimental Seeder Sample | SZ-3.6 (Astra) | Experimental Seeder Sample | SZ-3.6 (Astra) |
|---|---|---|---|---|
| Culture | Wheatgrass Burabay | Wheatgrass Burabay | Awnless bromegrass Akmola emerald | Awnless bromegrass Akmola emerald |
| Speed, km/h | 7.0 | 7.0 | 7.0 | 7.0 |
| Seeding rate, kg/ha actual | 8.61 | 8.81 | 13.72 | 14 |
| Sowing depth, cm | 4 | 4 | 4 | 4 |
| Sowing depth: (a) medium, cm | 4.11 | 4.07 | 4.08 | 4.04 |
| (b) standard deviation, ±cm | 0.2 | 0.39 | 0.25 | 0.44 |
| (c) variation coefficient, % | 4.81 | 9.76 | 6.06 | 10.95 |
| (d) seeds sown in the average actual layer depth and two adjacent layers, % | 91 | 86 | 90 | 84.3 |

**Table 2.** *Cont.*

| Name of Indicators | Experimental Seeder Sample | SZ-3.6 (Astra) | Experimental Seeder Sample | SZ-3.6 (Astra) |
|---|---|---|---|---|
| Seeds not sown into the soil, pieces/m$^2$ | No. | No. | No. | No. |
| Distributed plants: (a) quantity of plants in a 5 cm section, pieces | 4.9 | 4.8 | 5.5 | 5.6 |
| (b) standard deviation, ±pcs | 3.04 | 3.07 | 3.52 | 3.56 |
| (c) variation coefficient, % | 62.0 | 68.6 | 64 | 69.2 |

**Table 3.** The comparative indicators of the yield structure of wheatgrass sown on the experimental plot with the experimental seeder and sown on the control plot with the SZ 3.6 seeder (Astra).

| | Sprouted Plants, units/m$^2$ | The Height of the Plant, cm | Green Matter Yield, kg/ha |
|---|---|---|---|
| Seeder for non-flowing grass seed sowing | 293 | 85 | 52.989 |
| | 295 | 81 | 50.840 |
| | 315 | 83 | 55.628 |
| | 319 | 86 | 58.370 |
| | 320 | 79 | 53.787 |
| $\overline{X}$ | 308.4 | 82.8 | 54.323 |
| Seed drill SZ-3.6 (Astra) | 280 | 82 | 48.851 |
| | 317 | 83 | 55.981 |
| | 316 | 77 | 51.770 |
| | 295 | 79 | 49.585 |
| | 308 | 76 | 49.804 |
| $\overline{X}$ | 303.2 | 79.4 | 51.198 |

**Table 4.** Yield structure of awnless bromegrass on the experimental plot with the experimental model of a non-driven grass seed drill and on the control plot with seed drill SZ-3.6 (Astra).

| | Sprouted Plants, pcs/m$^2$ | The Height of the Plant, cm | Green Matter Yield, kg/ha |
|---|---|---|---|
| Seeder for non-flowing grass seed sowing | 310 | 105 | 70.761 |
| | 326 | 102 | 72.287 |
| | 290 | 103 | 64.935 |
| | 315 | 110 | 75.326 |
| | 324 | 108 | 76.070 |
| $\overline{X}$ | 313.0 | 105.6 | 71.876 |
| Seed drill SZ-3.6 (Astra) | 270 | 95 | 55.761 |
| | 305 | 102 | 67.630 |
| | 279 | 108 | 65.504 |
| | 290 | 102 | 64.304 |
| | 326 | 112 | 79.374 |
| $\overline{X}$ | 294.0 | 103.8 | 66.515 |

The Table 1 analysis shows that the germinating capacity of Burabay grass seeds was 90% when sown in the experimental plot and 86.44% in the control plot. The difference in the germinating capacity was 3.56%. The germination of awnless bromegrass seeds sown in the experimental plot was 89.0% and 86.15% in the control plot, and the increase in germination was 2.85%. An increase in the germinating ability of grass seeds is reached at the expense of the high quality of work of an experimental sowing unit and the seeding part of a prototype seeder. The experimental seeding unit does not damage the grass seed and creates a uniform flow of seeds.

**Table 5.** Experimental results for the energy assessment of the experimental non-flowing grass seed drill and the serial SZ-3.6 (Astra).

| Composition of the Unit | Sowing Depth, cm | Machine Speed, km/h | Theoretical Traction Resistance, kN | Traction Resistance, kN | Average Fuel Consumption, kg/hour | Average Slipping Ratio, % |
|---|---|---|---|---|---|---|
| HS1204 tractor with the experimental seeder | 2 | 5 | 3.11 | 3.1 | 13.88 | 19.37 |
| | | 7 | 3.13 | 3.15 | 13.75 | 22.12 |
| | | 9 | 3.26 | 3.4 | 13.67 | 24.78 |
| | 3 | 5 | 3.39 | 3.5 | 13.96 | 19.96 |
| | | 7 | 3.4 | 3.6 | 13.94 | 22.56 |
| | | 9 | 3.69 | 4.2 | 13.69 | 25.45 |
| | 4 | 5 | 3.66 | 3.7 | 14.05 | 20.54 |
| | | 7 | 3.68 | 4 | 14.04 | 23.13 |
| | | 9 | 4.15 | 4.6 | 14.35 | 25.31 |
| | 5 | 5 | 4.2 | 4.52 | 14.25 | 21.67 |
| | | 7 | 4.3 | 4.84 | 14.27 | 22.22 |
| | | 9 | 5.1 | 5.89 | 14.58 | 26.40 |
| HS1204 tractor with the series seeder | 2 | 5 | 3.48 | 3.53 | 15.4 | 21.40 |
| | | 7 | 3.49 | 3.59 | 15.26 | 24.47 |
| | | 9 | 3.5 | 3.87 | 15.20 | 27.45 |
| | 3 | 5 | 3.75 | 3.99 | 15.53 | 22.00 |
| | | 7 | 3.77 | 4.1 | 15.53 | 24.90 |
| | | 9 | 3.79 | 4.78 | 15.25 | 28.16 |
| | 4 | 5 | 4.02 | 4.22 | 15.66 | 22.61 |
| | | 7 | 4.04 | 4.56 | 15.68 | 25.48 |
| | | 9 | 4.07 | 5.24 | 16.09 | 27.83 |
| | 5 | 5 | 4.57 | 5.15 | 15.96 | 23.76 |
| | | 7 | 4.59 | 5.52 | 16.01 | 26.60 |
| | | 9 | 4.67 | 6.71 | 16.43 | 28.94 |

The analysis presented in Table 2 indicates that during the sowing of Burabay wheatgrass seeds, the seeding depth uniformity achieved by the experimental plot seeder is 4.81%, whereas the uniformity of the seeding depth on the control plot is 9.76%. Thus, the seeder from the experimental model for seeding non-flowing seeds surpasses the series seeder for seeding uniformity by 4.95%. On the experimental plot, the number of seeds sown to actual depth was 91.0% and was 86.0% on the control plot. Therefore, the experimental seeder outperforms the standard one by 5%. When sowing awnless bromegrass seeds "Akmolinsky emerald", the sowing depth uniformity of the seed was 6.06% on the experimental plot sown with an experimental seeder. On the other hand, on the control plot, the seeding depth uniformity was 10.95%. Consequently, the experimental seeder demonstrates an improvement of 4.89% in seeding depth compared to the standard seeder.

The analysis of the yield structure of wheatgrass (Table 3) showed that the quantity of germinated seeds in the experimental field was at 308.4 pcs/m$^2$, while in the control field it was at 303.2 pcs/m$^2$, resulting in 5.2 pcs/m$^2$. The average plant height on the experimental plot was 3.4 cm higher than the control plot counterparts (82.8 and 79.4 cm). Consequently, the Burabay yield rapeseed oil on the experimental plot sown with the experimental seeder was 54.323 cwt/ha and 51.198 cwt/ha on the control plot, resulting in a 3.125 cwt/ha increase in yield. Similar data were also obtained for awnless bromegrass "Akmola emerald" (Table 3).

The analysis revealed that the quantity of sprouted plants on the experimental plot exceeded the quantity of sprouted plants on the control plot (313 and 294 pcs/m$^2$). In addition, the height of the plant on the experimental plot was higher than on the control plot (105.6 and 103.8 cm). The yield of awnless bromegrass "Akmolinsky emerald" on the experimental plot sown with an experimental seed drill was 71.876 cwt/ha, while the control was 66.515 cwt/ha. Consequently, the increase in yield was 5.361 cwt/ha. Similar data were also obtained for awnless bromegrass "Akmola emerald" (Table 4).

The laboratory and field tests were conducted to assess the energy consumption of the experimental seed drill for seeding non-flowing grass and the serial SZ-3.6 (Astra). The results are shown in Table 5.

The results of analytical and laboratory and field studies of the experimental seeder for non-flowing grass seed and the serial seeder SZT-3.6 (Astra) [21] are presented in Figures 9–15. An analysis of Figure 9 shows that the resistance of the traction of the experimental seed drill increases with increasing seed placement depth. At the 2 cm placement depth, the traction resistance equals 3.4 kN, at 3 cm–4.2 kN, at 4 cm–4.6 kN, and at 6 cm–5.89 kN. Moreover, the experimental (Re) dependence between the traction resistance of the experimental seed drill and the depth of seeding at 9 km/h working speed exceeds the values of the theoretical (Rt) dependence of the traction resistance. This difference ranges from 4.2% at a seeding depth of 2 cm, escalating to 13.41% at a seeding depth of 5 cm.

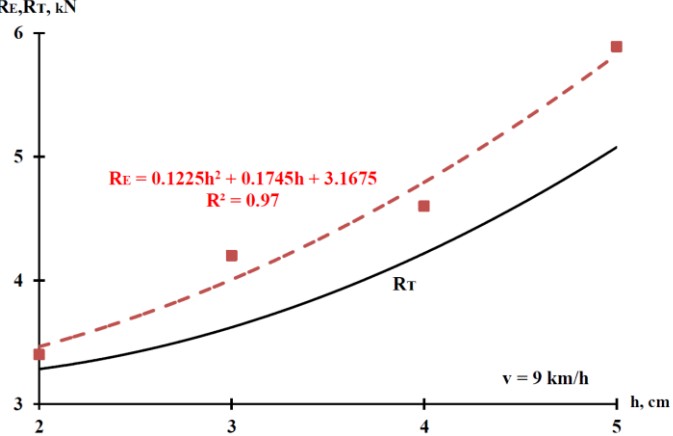

**Figure 9.** Experimental (Re) and theoretical (Rt) dependencies of draft resistance of the developed seeder on sowing depth at a working speed of 9 km/h.

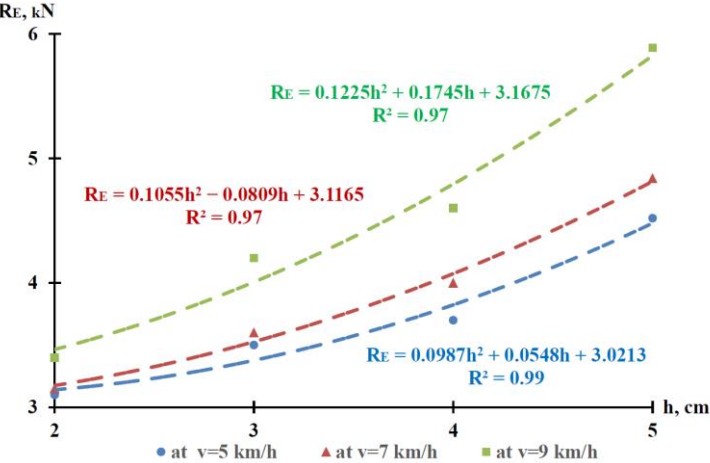

**Figure 10.** The experimental (Re) dependencies of draft resistance of the developed seeder on sowing depth at different aggregate speeds.

The experimental (Re) dependencies of the traction resistance of the experimental seeder on the depth of seed placement at different unit speeds are shown in Figures 10 and 11. The increase in traction resistance on the depth of seed placement is observed at various operating speeds of the sowing unit. The increase in the sowing depth causes the increase in traction resistance of the developed seed drill.

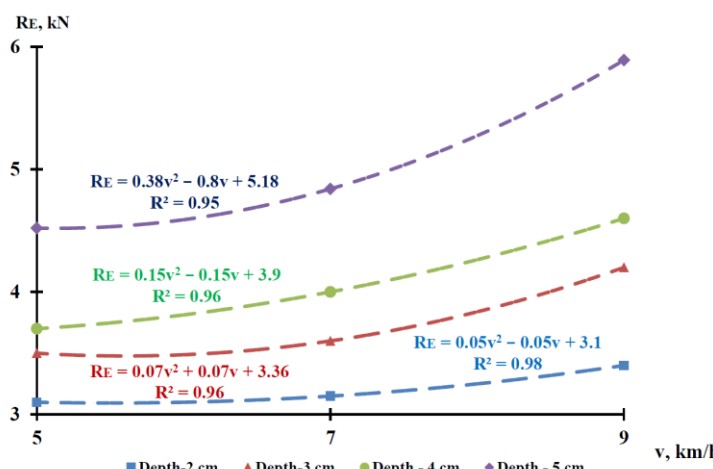

**Figure 11.** The experimental (Re) dependencies of draft resistance of the developed seeder on the aggregate speed at different sowing depths.

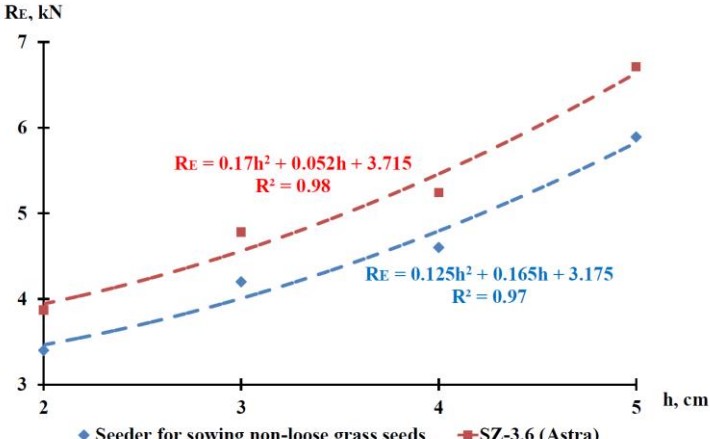

**Figure 12.** The experimental (Re) dependencies of draft resistance of the developed and standard seeder on sowing depth at a speed of 9 km/h.

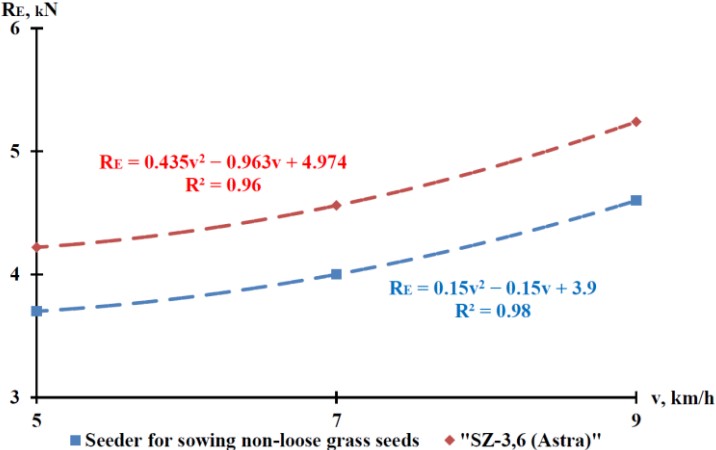

**Figure 13.** The experimental (Re) dependencies of traction resistance of the developed and serial seeder on the unit speed at a sowing depth of 4 cm.

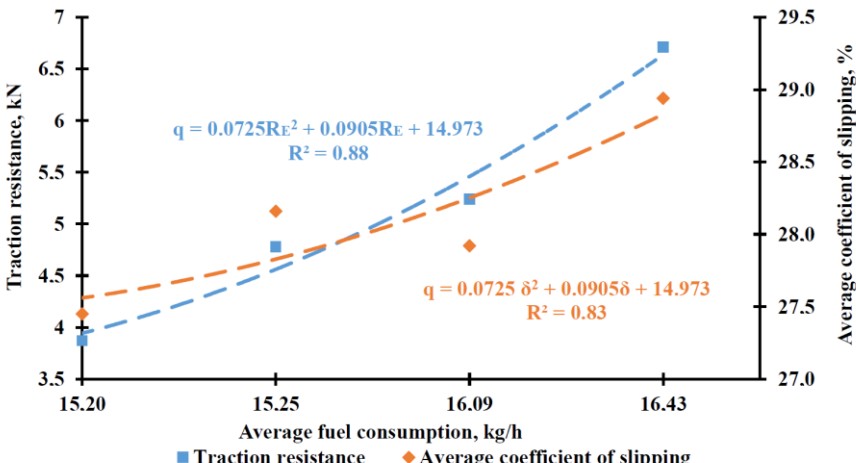

**Figure 14.** The average fuel consumption during the sowing of non-flowing grass seed on the traction resistance of the developed seeder and average tractor slip coefficient at a working speed of 9 km/h.

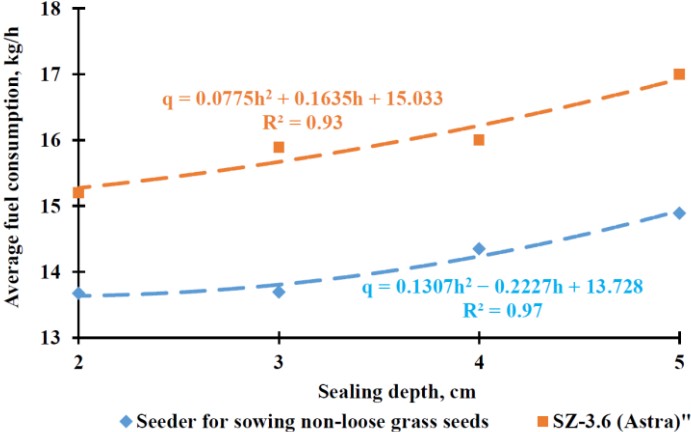

**Figure 15.** The dependencies of average fuel consumption on the sowing depth of developed and serial seeders at a working speed of 9 km/h.

An analysis shows that the experimental seeder traction resistance is lower than the traction resistance of the serial one by around 12.2% (Figure 12).

The analysis of Figure 13 shows that the traction resistance of the experimental seed drill is lower than the serial one by around 12.3%.

The increase in traction resistance of the developed seed drill results in heightened fuel consumption. A similar relationship is observed between fuel consumption and the coefficient of tractor slippage (Figure 14).

Figure 15 illustrates that the average fuel consumption rises along with the increase in the sowing depth. The difference of the average fuel consumption between the developed and the serial seeder increases from 8 to 10.2%. The same correlation exists between fuel consumption and the coefficient of tractor slippage.

For each type of seed drills for sowing seeds with low flowability and increased connectivity, a large number of both mechanical and pneumatic sowing systems with various technological techniques have been developed to ensure strict norms. For sowing such valuable grasses as rump, granary, wheatgrass, and fescue (sowing with diluents), seeders SZS-2.1, SZT-3.6, and SZP-3.6B have been developed. The North Caucasus Research Institute of Mechanization and Electrification of Agriculture has developed seeders using replaceable coulter blocks and vibro-discrete seeding devices that allow for the sowing of the entire range of agricultural crops [48]. Currently, among all the approaches aimed at improving (creating) hayfields and pastures, the method involving root cultivation has

gained the widest acceptance. This technique involves sowing highly valuable grasses into the existing vegetation using specialized combined machinery suitable for both compact and loose types of grasses. [49]. However, all developed machines are adapted to their soil and climatic conditions, which makes it impossible to compare the results with the obtained data.

## 4. Conclusions

The results of the economic tests established that:

- The germinating capacity of Burabay wheat grass seeds on the plot sown by the experimental seeder was 3.56% higher than the germinating capacity of seeds on the control plot. The improvement of grass seed germination was caused by the high quality of the experimental seeding unit and the seeding part of the prototype seeder.
- The experimental seeder outperformed the standard one by 4.95% in sowing wheat grass in terms of uniform seed placement depth.
- The yield increased by 5.361 kg/ha on the test plot sown with a non-drifted grass seed drill compared to the control plot.
- The traction resistance of the experimental seeder was 12.3% lower than that of the series seeder.

According to the technical characteristics of the prototype seeder, the operating efficiency was calculated for sowing non-flowing grass seeds. The cost-effectiveness of the experimental seeder sample was assessed using costs incurred during sowing compared to the increase in yields and the reduction in fuel and lubricant costs. The annual financial effect of the developed seeder due to an increase in grass yields and a reduction in fuel costs was 7714 USD/ha per year.

**Author Contributions:** Conceptualization, M.A. and S.N.; methodology, M.A.; software, T.T.; formal analysis, B.K. and M.B.; investigation, T.T., K.V. and K.U.; data curation, M.B. and T.T.; writing—original draft preparation, M.A., M.B. and S.N.; writing—review and editing, M.A., M.B., B.K. and S.N.; visualization, B.K. and M.B., supervision, M.A., M.B. and S.N.; project administration, M.A. All authors have read and agreed to the published version of the manuscript.

**Funding:** This research was funded by the Ministry of Science and Higher Education of the Republic of Kazakhstan, grant number AR19676894.

**Institutional Review Board Statement:** Not applicable.

**Data Availability Statement:** The data presented in this study are available upon request from the corresponding author.

**Acknowledgments:** The authors thank Jan Pawlik for proofreading the article.

**Conflicts of Interest:** The authors declare no conflict of interest.

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
