# Peer review of "Experimental Field Tests of the Suitability of a New Seeder for the Soils of Northern Kazakhstan"

_agriculture, doi:10.3390/agriculture13091687_

Round 1

Reviewer 1 Report

Reviewer comments

Some of the content still needs to be further improved. The proposed changes are as follows.

Point 1: Missing period on line 23

Point 2: Abstract needs to be reorganized. The abstract explains the progress brought about by the experiment, but does not explain the contribution made by this article in these improvements. There is no description of the new seeder, that is, the focus of this article is not included in the abstract.

Point 3: It is best to reduce the number of keywords to 6.

Point 4: The first paragraph of the introduction needs to be appropriately condensed to more clearly present the current problems and the necessity of this study.

Point 5: This article introduces a new type of seeder, but the introduction chapter extensively introduces the current high demand for new seeders. However, it does not introduce the current research on such seeders or the specific differences from the new seeders studied in this study.

Point 6: Line 157-160, There are too many “:”.

Point 7: There is a lack of discussion section, which should include a comparative analysis of the technical parameters or experimental results of this study with existing research to identify the innovative aspects of this article. If possible, please list a separate chapter for the discussion.

Point 8: Line 442-451, The first letter of each paragraph is not capitalized.

Point 9: Line 157-165, Line 173-180, How to determine the key component parameters of the machine? Please provide the selection basis for some important parameters to demonstrate that selecting this parameter can achieve better sowing results.

The English expression of this article is generally okay, some grammar and punctuation marks need to be checked again, please check the full text.

Author Response

Dear Reviewer,

Thank you very much for taking the time to read our manuscript thoroughly and make recommendations for its correction and improvement. We have read the comments carefully and have referred to all your comments.

Remark 1: Missing period on line 23

Answer: It was corrected.

Remark 2: Abstract needs to be reorganized. The abstract explains the progress brought about by the experiment, but does not explain the contribution made by this article in these improvements. There is no description of the new seeder, that is, the focus of this article is not included in the abstract.

Answer: Thank you for your comments. The entire abstract was completely replaced, expanded and the novelty of the developed working bodies and their design parameters were described.

Remark 3: It is best to reduce the number of keywords to 6.

Answer: Thank you for your comments. The number of keywords was reduced to 6.

Remark 4: The first paragraph of the introduction needs to be appropriately condensed to more clearly present the current problems and the necessity of this study.

Answer: Thank you very much for the suggestions to improve the article. The first paragraph of the introduction was completely removed. The repetitions about the herbs wheatgrass and rump were also removed. In the introduction, it is written in detail with links about the existing seeders imported to Kazakhstan. Kazakhstan has a lot of purchased equipment from abroad, as a rule they do not pass acceptance tests, the purchased machines are not adapted to the soil and climatic conditions of northern and central Kazakhstan. They have uneven sowing of difficult-to-flow seeds of forage grasses, expensive in cost, medium and small agricultural producers, which are much more than large ones, cannot purchase them. Kazakhstan has become dependent on imports, it does not produce its own equipment. In this connection, we offer our machine, which is designed for our soil and climatic conditions, at a much cheaper price and economical to use.

Remark 5: This article introduces a new type of seeder, but the introduction chapter extensively introduces the current high demand for new seeders. However, it does not introduce the current research on such seeders or the specific differences from the new seeders studied in this study.

Answer: Thank you very much for your comments. I'm giving clarifications. Kazakhstan is a livestock country and feed production requires a lot of attention. In order to have a balance between the number of animals and pastures, it is necessary to sow high-yielding varieties of fodder crops. High technologies require high-tech machines. Today, all equipment is imported to Kazakhstan from abroad. We need our own machines adapted to our weather and soil conditions and, of course, affordable.

Among the imported equipment there are many good, high-tech machines, but they are all very large metal-intensive, which means they are expensive, high fuel consumption, expensive to maintain, when there are no spare parts, the machines are idle. Our prototype planter for small-seeded, non-flowing grass seeds, is fully developed by our research team and adapted to the soil and climatic conditions of our region. This planter has new round hoppers, a new frame 4 meters wide, specially designed for a certain number of disc coulters with new packers and a new seed meter. All new elements are published in other articles (there are links to articles below) and a patent for the developed seeder has been received (Patent for invention (19) KZ (13) B (11) 35326 "Grass seeder"; dated 10/22/2021, bull. No. 42 https://bulletinofscience.kazatu.edu.kz/index.php/bulletinofscience/article/download/1072/849 ).

In connection with this, we cannot repeat the calculations and theoretical calculations in this article. In this article, at every mention, there are all references, both to published articles and to patents. In this article, we wanted to show the results of laboratory and field tests of the seeder we have already created and what results we got. A sowing device designed specifically for poorly flowing grass seeds, using the example of wheatgrass and awnless brome. These are high-yielding fodder crops that are zoned and allowed for production in Northern Kazakhstan, resistant to lodging, drought-resistant and seeds do not crumble.

Our developed seeder is much cheaper than purchased machines, the diameter of the developed disc coulters can be standard, but the parameters between the discs are α = 10°, the vanishing point of the disc is β = 40°, the coulter angle is α= 32° precisely for our soil and climatic conditions and new press wheels, of a configuration that presses the grass seeds in such a way that they are not strongly pressed into the soil, but have close contact with the soil. Theoretical studies are in the articles, in the links provided. Such seeders would be affordable for small and medium-sized agricultural producers.

Remark 6: Line 157-160, There are too many “:”.

Answer: It was corrected

Remark 7: There is a lack of discussion section, which should include a comparative analysis of the technical parameters or experimental results of this study with existing research to identify the innovative aspects of this article. If possible, please list a separate chapter for the discussion.

Answer: Thanks a lot for the suggestions. In the last section, where an analysis of laboratory and field studies is given, in our opinion, a detailed discussion of the results obtained, characterizing the operation of the proposed seeder, was carried out. In each study, there is data on a serial seeder in comparison with our developed seeder, and all the obtained parameters in a comparative analysis are shown both in tables and in graphs. It is impossible to compare two machines only by the technical parameters of the sowing unit or only by disc coulters, or only by leveling wheels in the field. The operation of the machine and the quality of the production process are evaluated in conjunction with the sowing of seeds, the distribution of seeds, and the rolling of seeds. Comparison with existing planters is the analysis we talked about in the introduction. For existing or, as we say, for imported seeders, when sowing difficult-to-flow seeds, unloading occurs, that is, the seeds are entangled with each other, a lump has formed and does not wake up from the sowing machine and there is no further movement in the seed tube. We say that in our sowing machine the seeds do not stick together, there is a uniform sowing. Same with coulters. That is, the final result of the operation of the machine is compared, and not individual working bodies in action. The discussion and analysis of the obtained results is given in detail in this section. In this connection, we are very grateful to you for your work and for giving our article a lot of attention.

Remark 8: Line 442-451, The first letter of each paragraph is not capitalized.

Answer: It was corrected

Remark 9: Line 157-165, Line 173-180, How to determine the key component parameters of the machine? Please provide the selection basis for some important parameters to demonstrate that selecting this parameter can achieve better sowing results.

Answer: Thank you for your recommendation. The design parameters of the developed working bodies and their novelty of the seeder are confirmed by patents. Calculations for the dimensions of the mechanical structure of the working bodies are given in patents and articles listed below.

  1. Patent for invention (19) KZ (13) B (11) 35326 “Grass seeder”; dated 22.10.2021, bul. No. 42 https://bulletinofscience.kazatu.edu.kz/index.php/bulletinofscience/article/download/1072/849

Articles published in journals:

  1. Aduov M.A., Nukusheva S.A., Volodya K. Analysis of the operation process of the sieve apparatus for non-flowing seeds of fodder crops. Bulletin of Science of the Kazakh Agrotechnical University. S. Seifullina, Astana, 2018, No. 2 (97), p. 146-151. ISSN2079-939X http://bulletinofscience.kazatu.kz/assets/i/journals/2(97)2018/%D0%90%D0%B4%D1%83%D0%BE%D0%B2.%D0%9C.%D0 %90.pdf
  2. Aduov M.A., Nukusheva S.A., Kuanyshova A.Zh., Volodya K. Results of experimental studies of the sowing apparatus for non-flowing seeds of fodder crops. Bulletin of Science of the Kazakh National Agrarian University "Research, results" No. 1 2018, p.392-400. ISSN2304-3334-03, https://izdenister.kaznau.kz/files/full/2018_1.pdf
  3. Aduov M.A., Nukusheva S.A., Volodya K. Substantiation of the main design and technological parameters of the closing part of the seeder for sowing grass. Bulletin of Science of KATU named after S. Seifullin, - 2022, - No. 2 (113), Part 2 - P. 66-77. DOI: https://doi.org/10.51452/kazatu.2022.2(113).1072
  4. Aduov M.A., Nukusheva S.A., Tulegenov T.K., Isenov L.G., Volodya K. Substantiation of the design parameters of an individual press roller of a seeder for sowing grass. Bulletin of Science of KATU named after S. Seifullin, - 2022, - No. 3 (114), Part 1 - S. 200-210. DOI: https://bulletinofscience.kazatu.edu.kz
  5. Innovative patent of the Republic of Kazakhstan No. 31106. (19) KZ (13) A4 (11) 31106. Seeding machine, Bulletin No. 5 of 05/16/2016. https://kazatu.edu.kz/assets/i/science/sf13_shmash_110.pdf

Reviewer 2 Report

In this study, comprehensive tests were conducted on an experimental seeder with new seed drills and seeding units. Wheatgrass varieties “Burabay” and awnless brome “Akmola emerald” were sown. To solve significant sowing problems can be used in the agro-industrial sector in the development of the 25 livestock sector, regardless of the country or region, when improving the forage base in similar soil 26 and climatic zones.The economic efficiency of the experimental sample of the seeder was obtained from the financial costs incurred during sowing in comparison with an increase in yield and a decrease in the cost of fuels and lubricants.

I think there are several questions that need to be answered.

1. A seeder for non-flowing grass a seed has been developed, which is shown in Figure 1. However, this machine is suspended and did not land. As shown in this picture, I think this photo should be a product promotion photo. If this photo is not an original mechanical creation by the author, the author needs to add the source of the photo. If it's an existing product, I don't think the author would be suitable to say they developed it themselves.

2. In Figure 4, there is clearly an additional symbol at position “-”.

Overall, the research work of this article is still very detailed and comprehensive. According to the technical characteristics of the prototype seeder, the operating efficiency was calculated for sowing non-friable grass seeds. The cost-effectiveness of the experimental seeder sample was assessed using costs incurred during sowing compared  to the increase in yields and the reduction in fuel and lubricant costs.

Author Response

Dear Reviewer,

Thank you very much for taking the time to read our manuscript thoroughly and make recommendations for its correction and improvement. We have read the comments carefully and have referred to all your comments.

Remark 1: A seeder for non-flowing grass a seed has been developed, which is shown in Figure 1. However, this machine is suspended and did not land. As shown in this picture, I think this photo should be a product promotion photo. If this photo is not an original mechanical creation by the author, the author needs to add the source of the photo. If it's an existing product, I don't think the author would be suitable to say they developed it themselves.

Answer: Thank you for your remark, in fact the drawing is not very clear. The picture was replaced.

Remark 2: In Figure 4, there is clearly an additional symbol at position “-”.

Answer: It was corrected.

Reviewer 3 Report

This study attempted to solve significant sowing problems in Kazakhstan, which is important and can be of high practical value. The paper conducted on an experimental seeder with new seed drills and seeding units and designed some experiments to ensure a quality sowing process of hardy and non-drilled grass seeds. Research showed the higher efficiency of the experimental seeder with seeding units and sowing parts compared to a serial seeder in terms of agricultural performance. This is a subject which is a good fit for this journal.

 Authors could improve part of the manuscript.  Here are some specific recommendations:

 Title: It is too long.

Keywords: There are many keywords that can be simplified.

1. Introduction:

(1)The logic in the introduction section is somewhat confusing. The paper did not clearly explain the background, significance, and current key issues of the seeding machine in the introduction.

(2)The theme of this article is to introduce a new type of seeder and seeding device suitable for Wheatgrass Varieties Burabay and Awnless Brome Akmola Emerald grass seeds with poor fluidity. Therefore, it should include a brief introduction of the grass variety. However, there is a lot of information about grass, so it is recommended to simplify this part.

(3)Suggest adding an introduction to the research status of relevant seeders and analyzing the problems that arise during their work process.

 2. Materials and Methods

(1)There are more contents describing the structure and dimensions, but the introduction of the working principle is not clear. The writing of the entire mechanical part is more like the writing of a patent.

(2)Change the positions of symbols 5 and 6 in Figure 1.

(3)The determination of mechanical structure dimensions is directly given and lacks theoretical analyses and calculation. The basis for designing mechanical structure dimensions should be added, which is not only a design difficulty but also an innovation point that this paper aims to address. The determination of size cannot be separated from the physical properties of the seed, so this section should provide important physical properties and photos of the material (Wheatgrass Varieties Burabay and Awnless Brome Akmola Emerald grass seeds).

(4)The paper mentions that the control unit for sowing is important for sowing quality, but only the installation position of the sensor is provided without further description and explanation in the paper.

(5)Suggest using tables for experimental design, which may be clearer.

3. Results and Discussion

(1)Tables 2 and 3 can be placed together for comparative analysis, while Tables 4 and 5 can be placed together too.

(2)Is the data in Table 6 the same as the data in Figures 7-13? If it's the same, it's repetition. It is suggested to change Table 6 into design of experiments and put it in the Materials and Methods section of the paper.

(3)What is the reason for the large standard deviation and coefficient of variation in the last two rows of Table 1? Need to explain.

Language: The English language of the entire article is inaccurate, and specialized vocabulary needs to be standardized. 

Author Response

Dear Reviewer,

Thank you very much for taking the time to read our manuscript thoroughly and make recommendations for its correction and improvement. We have read the comments carefully and have referred to all your comments.

Remark 1: Title: It is too long

Answer: We agree that the title is long. We have shortened the title of the article and propose the following one: Experimental Field Tests of the Suitability of the New Seeder for the Soils of Northern Kazakhstan

Remark 2: Keywords: There are many keywords that can be simplified.

Answer: It was corrected.

Remark 3: The logic in the introduction section is somewhat confusing. The paper did not clearly explain the background, significance, and current key issues of the seeding machine in the introduction.

Answer: Thank you for the remark. It was corrected in the article.

Remark 4: The theme of this article is to introduce a new type of seeder and seeding device suitable for Wheatgrass Varieties Burabay and Awnless Brome Akmola Emerald grass seeds with poor fluidity. Therefore, it should include a brief introduction of the grass variety. However, there is a lot of information about grass, so it is recommended to simplify this part.

Answer: We agree with recommendations. The description of herbs has been reduced; these are widespread crops in the Northern region of Kazakhstan.

Remark 5: Suggest adding an introduction to the research status of relevant seeders and analyzing the problems that arise during their work process.

Answer: Thank you for the remark, we corrected it in the introduction, and added some relevant information about developed working parts.

Remark 6: There are more contents describing the structure and dimensions, but the introduction of the working principle is not clear. The writing of the entire mechanical part is more like the writing of a patent.

Answer: Thanks for the recommendation. We have improved the description of working parts.

Remark 7: Change the positions of symbols 5 and 6 in Figure 1.

Answer: Figure 1 replaced with another picture

Remark 8: The determination of mechanical structure dimensions is directly given and lacks theoretical analyses and calculation. The basis for designing mechanical structure dimensions should be added, which is not only a design difficulty but also an innovation point that this paper aims to address. The determination of size cannot be separated from the physical properties of the seed, so this section should provide important physical properties and photos of the material (Wheatgrass Varieties Burabay and Awnless Brome Akmola Emerald grass seeds).

Answer: Thanks for the remark. The design parameters of the developed working bodies and their novelty of the seeder are confirmed by patents. Calculations for the dimensions of the mechanical structure of the working bodies are given in patents and articles listed below.

  1. Patent for invention (19) KZ (13) B (11) 35326 “Grass seeder”; dated 22.10.2021, bul. No. 42 https://bulletinofscience.kazatu.edu.kz/index.php/bulletinofscience/article/download/1072/849

Articles published in journals:

  1. Aduov M.A., Nukusheva S.A., Volodya K. Analysis of the operation process of the sieve apparatus for non-flowing seeds of fodder crops. Bulletin of Science of the Kazakh Agrotechnical University. S. Seifullina, Astana, 2018, No. 2 (97), p. 146-151. ISSN2079-939X http://bulletinofscience.kazatu.kz/assets/i/journals/2(97)2018/%D0%90%D0%B4%D1%83%D0%BE%D0%B2.%D0%9C.%D0 %90.pdf
  2. Aduov M.A., Nukusheva S.A., Kuanyshova A.Zh., Volodya K. Results of experimental studies of the sowing apparatus for non-flowing seeds of fodder crops. Bulletin of Science of the Kazakh National Agrarian University "Research, results" No. 1 2018, p.392-400. ISSN2304-3334-03, https://izdenister.kaznau.kz/files/full/2018_1.pdf
  3. Aduov M.A., Nukusheva S.A., Volodya K. Substantiation of the main design and technological parameters of the closing part of the seeder for sowing grass. Bulletin of Science of KATU named after S. Seifullin, - 2022, - No. 2 (113), Part 2 - P. 66-77. DOI: https://doi.org/10.51452/kazatu.2022.2(113).1072
  4. Aduov M.A., Nukusheva S.A., Tulegenov T.K., Isenov L.G., Volodya K. Substantiation of the design parameters of an individual press roller of a seeder for sowing grass. Bulletin of Science of KATU named after S. Seifullin, - 2022, - No. 3 (114), Part 1 - S. 200-210. DOI: https://bulletinofscience.kazatu.edu.kz
  5. Innovative patent of the Republic of Kazakhstan No. 31106. (19) KZ (13) A4 (11) 31106. Seeding machine, Bulletin No. 5 of 05/16/2016. https://kazatu.edu.kz/assets/i/science/sf13_shmash_110.pdf

Remark 9: The paper mentions that the control unit for sowing is important for sowing quality, but only the installation position of the sensor is provided without further description and explanation in the paper.

Answer: Thank you for your mention. In this article, we are only talking about the analysis of the quality of the seeder, where we only mention that the seeder is smart and works with a control unit. If we talk about the control unit of the seeding system, this will be another article.

Remark 10: Suggest using tables for experimental design, which may be clearer.

Answer: Thanks for the suggestion. The tables given in this article show all the results obtained during the testing of the seeder. All of them are of a comparative nature with a serial seeder and with a specific comparative analysis.

Remark 11: Tables 2 and 3 can be placed together for comparative analysis, while Tables 4 and 5 can be placed together too.

Answer: Thanks for the advice, tables 2 and 3; 4 and 5 were combined

Remark 12: Is the data in Table 6 the same as the data in Figures 7-13? If it's the same, it's repetition. It is suggested to change Table 6 into design of experiments and put it in the Materials and Methods section of the paper.

Answer: Thanks for the recommendation. Table 6 is not a repetition of graphs, these are the results of experiments on the energy assessment of an experimental seeder, according to the meaning and logic, this we belive table should be in the discussion section, before the graphs.

Remark 13: What is the reason for the large standard deviation and coefficient of variation in the last two rows of Table 1? Need to explain.

Answer: Thank you for your remark. The table shows data that, according to statistics, is an acceptable percentage of the introduced coefficient of variation, i.e. 60-70% is the standard deviation of the seed placement depth, expressed as a percentage of the arithmetic mean given. In our case, this percentage is, if when sowing Burabay grass with an experimental seeder, the coefficient is 62.0%, and when sowing with a serial seeder 68.6%, then the coefficient of variation in the depth of seeding is improved by 5.4%, respectively, if 69.2% when sowing with a serial and experimental seeder 65%, then the coefficient of variation in the depth of seed placement is 5.2%, i.e. we say that seed distribution by planting depth improved by 5.4% and by 5.2%.

Remark 14: The English language of the entire article is inaccurate, and specialized vocabulary needs to be standardized. 

Answer: It was done

Round 2

Reviewer 1 Report

In the discussion section, it may be better to compare the results of this study with the latest research related to this study.  But, I think this article can be accepted. 

Author Response

Remark 1.

In the discussion section, it may be better to compare the results of this study with the latest research related to this study.  But, I think this article can be accepted. 

Answer:

Dear reviewer, thank you very much for the recommendation. In the discussion section, we have add some development in comparisons with other actual works.

Reviewer 3 Report

The language of the paper requires professional assistance in editing.

The research background and current research status still need to be improved, and the logic of the following questions needs to be explained clearly.

What are the problems faced by sowing grass seeds mentioned in the paper? What does the grass seed look like (photos and parameters can be provided)? Why are there problems with sowing? What is the solution principle of the new machine designed in the paper? The paper needs to be reorganized and explained these clearly.

The conclusion of 498-507 cannot be directly seen from Table 2.

The conclusions of 521-533 cannot be directly seen from Table 3.

The picture in Figure 6 is difficult to compare which is good and which is bad. Can you supplement the comparison photos of seedling emergence after germination, the middle stage photos, and the comparison photos before the final harvest? If do this this paper will be more powerful and rich.

Author Response

Dear Reviewer!

Thank you so much for all your remarks!

Remark 1.

The language of the paper requires professional assistance in editing.

Answer: The article was professionally proofread.

Remark 2.

The research background and current research status still need to be improved, and the logic of the following questions needs to be explained clearly.

Answer: Thank you for the remark. The article was improved.

Remark 3.

What are the problems faced by sowing grass seeds mentioned in the paper? What does the grass seed look like (photos and parameters can be provided)? Why are there problems with sowing? What is the solution principle of the new machine designed in the paper? The paper needs to be reorganized and explained these clearly.

Answer: Thank you for the remark. The article was suplemented.

Remark 4.

The conclusion of 498-507 cannot be directly seen from Table 2.

The conclusions of 521-533 cannot be directly seen from Table 3.

Answer: Tables were suplemented with the average data, which corresponds to the above analyses.

Remark 5.

The picture in Figure 6 is difficult to compare which is good and which is bad. Can you supplement the comparison photos of seedling emergence after germination, the middle stage photos, and the comparison photos before the final harvest? If do this this paper will be more powerful and rich.

Answer: The images of phenological observations in the results and discussion section were added.